# Cytosine modifications exhibit circadian oscillations that are involved in epigenetic diversity and aging

Gabriel Oh[1], Sasha Ebrahimi[1], Matthew Carlucci[1], Aiping Zhang[1], Akhil Nair[1], Daniel E. Groot[1], Viviane Labrie[1,2], Peixin Jia[1], Edward S. Oh[1], Richie H. Jeremian[1], Miki Susic[1], Tenjin C. Shrestha[3,4], Martin R. Ralph[4], Juozas Gordevičius[5,6], Karolis Koncevičius[5,6] & Art Petronis[1,5]

Circadian rhythmicity governs a remarkable array of fundamental biological functions and is mediated by cyclical transcriptomic and proteomic activities. Epigenetic factors are also involved in this circadian machinery; however, despite extensive efforts, detection and characterization of circadian cytosine modifications at the nucleotide level have remained elusive. In this study, we report that a large proportion of epigenetically variable cytosines show a circadian pattern in their modification status in mice. Importantly, the cytosines with circadian epigenetic oscillations significantly overlap with the cytosines exhibiting age-related changes in their modification status. Our findings suggest that evolutionary advantageous processes such as circadian rhythmicity can also contribute to an organism's deterioration.

[1] The Krembil Family Epigenetics Laboratory, The Campbell Family Mental Health Research Institute, Centre for Addiction and Mental Health, Toronto, ON M5T 1R8, Canada. [2] Center for Neurodegenerative Science, Van Andel Research Institute, Grand Rapids, MI 49503, USA. [3] Department of Cell and Systems Biology, University of Toronto, Toronto, ON M5S 3G5, Canada. [4] Department of Psychology, University of Toronto, Toronto, ON M5S 3G5, Canada. [5] Institute of Biotechnology, Life Sciences Center, Vilnius University, Vilnius, LT-10257, Lithuania. [6] Institute of Data Science and Digital Technologies, Vilnius University, Vilnius, LT-08663, Lithuania. Gabriel Oh and Sasha Ebrahimi contributed equally to this work. Correspondence and requests for materials should be addressed to A.P. (email: Art.Petronis@camh.ca)

Circadian rhythmicity is one of the oldest evolutionary adaptations to day and night cycles. It regulates a wide spectrum of biological phenomena, from temperature-dependent fluctuations in biochemical reaction rates of prokaryotes, to sleep-wake cycles and higher-order behaviors in multicellular organisms[1]. Disruptions of circadian rhythms have been linked to human morbidities, including cancer, mood, and neurodegenerative diseases[2,3]. Relatedly, numerous studies have shown an association between circadian disruption and aging. For instance, in older rodents, circadian regulation becomes weaker, whereas mice deficient in key circadian genes have shorter life-spans[4]. The cause-effect relationship and molecular mechanisms of this association are yet to be uncovered.

The cell-autonomous circadian clock consists of a series of transcription factors and regulators that coordinate feedback loops. In mammals, the clock circadian regulator (CLOCK) transcription factor forms a heterodimer with the aryl hydrocarbon receptor nuclear translocator-like protein (ARNTL, also known as BMAL1). This complex binds to the E-box response elements to regulate expression of clock controlled genes[5,6]. This set of activated genes includes *Period* (*PER1*, *2*, and *3*) and *Cryptochrome* (*CRY1* and *CRY2*), which initiate a negative feedback in this pathway. Circadian-regulated genes are ubiquitous and partially tissue specific: up to 55% of protein-coding genes exhibit circadian transcriptional oscillations in at least 1 of 12 mouse tissues[7]. A number of post-translational histone modifiers also mediate circadian regulation. CLOCK, for instance, functions as an acetyltransferase of histone H3 at K9 and K14 positions and interacts with other histone acetyltransferases[8]. Similarly, other circadian factors, directly or through formation of complexes with other enzymes, control histone deacetylation[9] and methylation[10]. Genome-wide chromatin immunoprecipitation experiments have demonstrated coordinated circadian oscillations of histone H3K4 trimethylation and H3K9/H3K27 acetylation at transcription start sites of expressed genes, as well as H3K4 monomethylation and H3K27 acetylation at enhancers[11]. Contrary to the strong evidence for histone oscillations, the role of cytosine modification, mainly comprised of 5-methyl- and 5-hydroxymethylcytosines (hmCs), in the maintenance or modulation of the circadian clock is unclear. Even though cyclical changes with 24 h periodicity have been demonstrated in key elements of the cytosine modification machinery (such as DNA methyltransferases, Ten-eleven translocation enzymes, and global DNA methylation levels[12]),

dedicated studies have previously failed to detect evidence for robust 24 h periodicity of cytosine modification patterns in mice[13,14]. In humans, a study of postmortem brain samples revealed vestiges of circadian oscillations, which accounted for a small fraction (< 0.3%) of the total inter-individual cytosine modification variance[15].

In this study (Fig. 1), we demonstrate that cytosine modification profiles are changing in a circadian manner in the mouse liver and lung. Oscillating modified cytosines (osc-modCs) are more prevalent in both high expressing and circadian genes and were enriched for E-box motifs. We also find that osc-modCs were associated with the aging epigenome, where the amplitude of the oscillation correlated with the magnitude of the aging effect, implying common molecular mechanisms and shedding a new light on the proximal causes of aging.

## Results

**Circadian oscillations of modCs.** We investigated circadian oscillations of cytosine modification in 9-, 15-, and 25-month-old (mo) male mice ($n = 36$, 30, and 30, respectively) to represent an aging spectrum from adult to the very old. All mice were individually housed, with ad libitum access to food and water, and entrained on a 12 h light : 12 h dark cycle, where Zeitgeber time (ZT) 0 is light onset and ZT12 is light offset. The circadian entrainment of mice was verified using locomotor activity and messenger RNA profiles of two key circadian genes, *Per2* and *Arntl*[1] (Fig. 2a–f). Liver and lung tissues, which exhibit robust transcriptional oscillations and are frequently used in circadian studies[7], were collected every 2 h for at least 58 h starting at ZT0.

We targeted chromosome 7, which is relatively small and gene dense, to gain sufficient depth of sequencing. In order to gain precise measurements of oscillating cytosine modifications all experiments were performed in technical triplicates, and the cytosine modification measurements were filtered for sites that showed a greater degree of biological variation than technical variation (henceforth referred to as epigenetically variable cytosines (EVCs); see Methods for more details). We performed mapping of osc-modCs at the single nucleotide resolution using 10,696 bisulfite padlock probes[16], each targeting a unique 130-140 bp region of chr 7, followed by sequencing to an average mapped read depth of ~ 1,800 × per CpG (Supplementary Fig. 1). Out of the 37,249 targeted CpGs (Supplementary Data 1), we

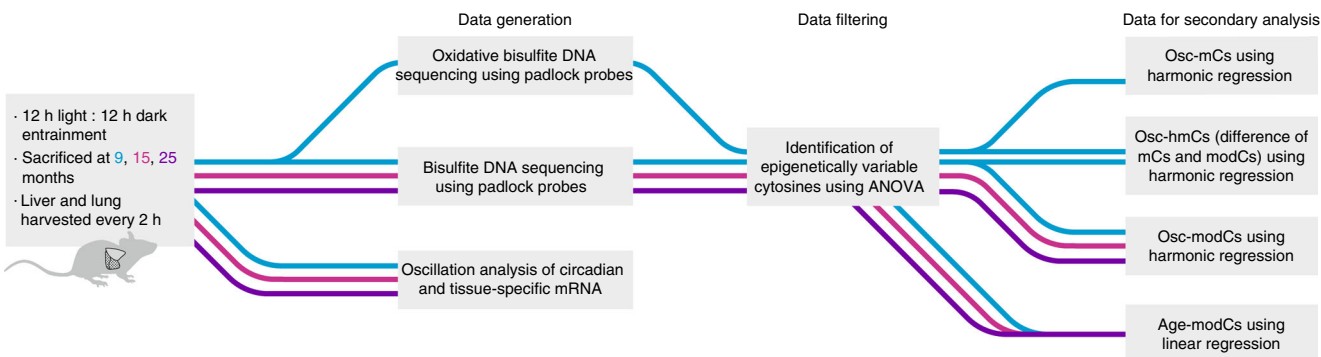

**Fig. 1** Experimental workflow summary. We investigated 9-, 15-, and 25-month-old (mo) male mice entrained on a 12 h light : 12 h dark cycle with ad libitum access to food. The circadian entrainment of mice was verified using locomotor activity and mRNA profiles of key circadian genes. The confounding effects of cellular heterogeneity were examined using cell-specific non-circadian mRNAs. All sequencing experiments were performed in technical triplicates. ANOVA was used to select for cytosines whose variation of modification densities were larger in the biological samples compared with technical replicates, which we refer to as epigenetically variable cytosines (EVCs). 5-Hydroxymethylcytosine densities were estimated by subtracting mC from modC densities on CpG sites that are intersected across EVCs. Aging and oscillating cytosines were identified using linear and harmonic regression models, respectively. Age-modC, age-correlated cytosine modifications; Osc-hmC, oscillating 5-hydroxymethylcytosines; Osc-mC, oscillating 5-methylcytosines; Osc-modC, oscillating modified cytosines

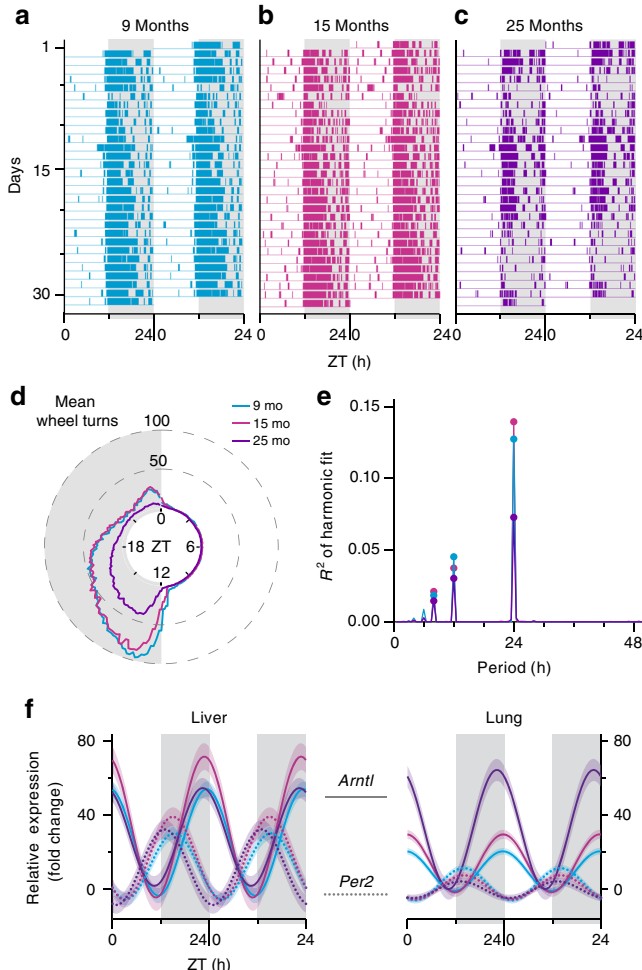

**Fig. 2** Locomotor activity and molecular markers verifying circadian entrainment of mice. **a-c** Representative actograms of individual mice from **a** 9-mo, **b** 15-mo, and **c** 25-mo cohorts. The colored bars represent wheel running activity over 30 days. The light and grey columns show periods of lights on and off. By convention, each line depicts 2 days of activity, with the second day appearing again in the line below (i.e., double-plotted). **d** The summarized activity of each age cohort measured as the mean number of turns for each mouse in the age group (measured in 6 min intervals over 30 days) as a function of ZT. **e** Spectral analysis of the actograms in each age cohort showing the $R^2$ of harmonic fits at varying periodicities (1 h intervals). **f** Twenty-four hours harmonic regression fits of relative mRNA levels of two antiphasic circadian genes, *Arntl* and *Per2*, in liver and lung tissues of all three age cohorts. The data are double plotted by convention. Shading around the regression lines represents the 95% confidence band. ZT, Zeitgeber time; mo, month-old

successfully captured ~ 28,000 CpGs, of which ~ 13,000 were epigenetically variable.

We plotted the average modification density within 1 Mb windows of chromosome 7 and observed a synchronized pattern of oscillation in the lung samples, whereas a less organized pattern was detected in the liver. Harmonic regression analysis, which can identify sinusoidal patterns with 24 h periodicity, was used to determine the periodic relationship between ZT and modification levels. It showed that chromosome-wide average modification oscillated with a 24 h periodicity in the lung ($p = 2.9 \times 10^{-7}$) but not in the liver ($p = 0.74$) (Fig. 3a-b). Principal component analysis (PCA) revealed that the oscillating first principal component (harmonic regression $p = 1.8 \times 10^{-6}$) explained 25% of the cytosine modification variance in the lung,

whereas PCA in the liver interestingly revealed oscillating principal components 2 and 3 (harmonic regression $p = 0.028$ and 0.015, respectively), which cumulatively explained 13% of the variance (Fig. 3c-d). Analysis of individual CpGs revealed osc-modCs in 8.2% (permuted $p = 0.046$) and 35.6% (permuted $p < 10^{-4}$) of EVCs in the liver and lung, respectively (Fig. 4a-f, Supplementary Fig. 2, and Supplementary Data 2-3). The mean amplitudes of oscillation were $3.2 \pm 1.8\%$ (mean $\pm$ SD) and $4.5 \pm 2.2\%$, with maximum amplitudes of 14% and 17%, in the liver and lung, respectively. Some osc-modCs were not tissue specific, as genomic positions of osc-modCs significantly overlapped between the liver and lung (odds ratio (OR) (95% confidence interval) = 2.0 (1.7-2.4); $p = 1.2 \times 10^{-20}$), and shared a similar acrophase (ZT when oscillation reaches its peak) with a median absolute acrophase difference of 2.1 h (permuted $p < 1.0 \times 10^{-4}$; Supplementary Fig. 3).

Consistent with the general trend of diminishing circadian effects with age[3,4], we observed a decreasing proportion of osc-modCs in the older animals. Compared with the 9-mo cohort, lung osc-modCs were reduced slightly in the 15-mo mice (28.1%) and dropped more dramatically in the 25-mo mice (13.9%; Supplementary Fig. 4 and Supplementary Data 4-5), and these effects were not influenced by differences in cohort sample sizes (Supplementary Table 1). Liver did not exhibit significant oscillations in the older age cohorts (Supplementary Fig. 4 and Supplementary Data 6-7). As only 9-mo mice showed consistently significant oscillations in both tissues, we focused primarily on this group for all subsequent circadian cytosine modification analyses.

Mechanistically, genuine osc-modCs must involve DNA demethylation, probably by the oxidation of 5-methylcytosine (mC) to hmC[17], followed by remethylation. Using oxidative bisulfite (oxBS) conversion[18] followed by sequencing using padlock probes, we detected mostly significant oscillations in both hmC and mC densities (permuted $p = 0.011$ and 0.095, for liver; $p = 0.012$ and $7.8 \times 10^{-3}$, for lung; Supplementary Fig. 5a-l and Supplementary Data 8-11). Although both types of oscillating modifications overlapped (OR = 6.3 (5.2–7.6) and 10.2 (5.7-18.4); $p = 9.2 \times 10^{-71}$ and $7.4 \times 10^{-16}$ in the liver and lung, respectively), their respective acrophases were in antiphase with one another (11.2 and 11.0 h median absolute acrophase difference between the two; permuted $p < 1 \times 10^{-4}$ and $5.4 \times 10^{-3}$ in the liver and lung, respectively; Supplementary Fig. 5m-p). This indicates coordinated timing of DNA demethylation and remethylation during the circadian cycle.

Circadian oscillations of cytosine modifications may be confounded by cyclic influx of white blood cells into solid tissues[19,20]. In such cases, changes in the proportions of cells may mimic osc-modCs due to contrasting epigenomes between penetrating blood cells and native tissue cells. If cell counts are a confounder, cell-specific non-circadian mRNAs should exhibit evidence for circadian oscillations. We investigated two non-circadian mRNAs for each of hepatocytes, pneumocytes, and macrophages. None of these cell-specific transcripts showed significant circadian oscillations (Supplementary Fig. 6 and Supplementary Table 2), suggesting that osc-modC were not simulated by changing cell counts.

**Osc-modCs in circadian genes and E-box motifs.** To test the functional association between osc-modCs and mRNA levels, we compared tissue-matched public circadian transcriptomic data sets[7] ($n = 72$ for both the liver and lung) with cytosine modification oscillations detected in each tissue. The oscillations for a given gene were summarized as the median coefficient of

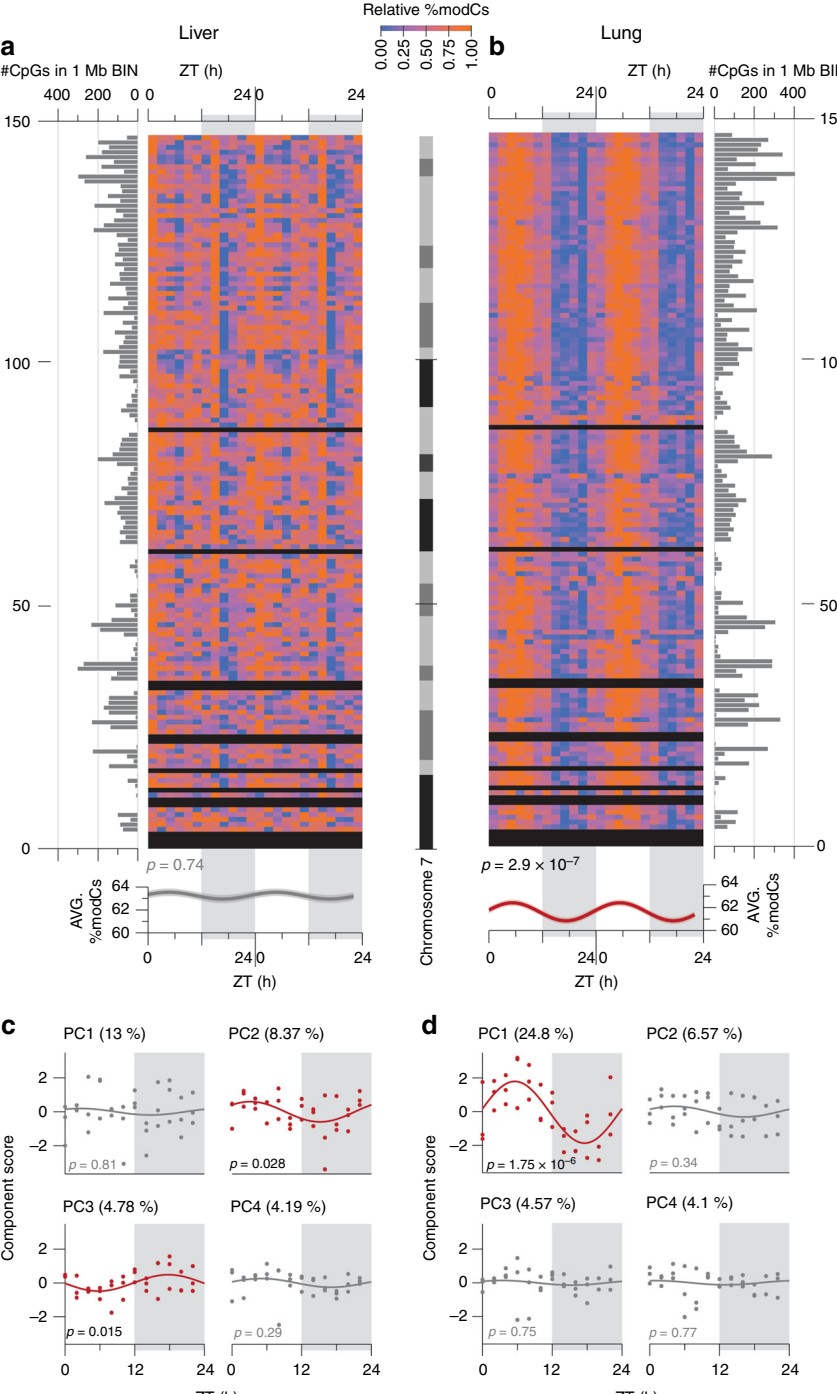

**Fig. 3** Global patterns of oscillations in the 9-mo mouse liver and lung. **a, b** A heatmap of the mean modification densities of epigenetically variable cytosines (EVCs) across chromosome 7 of **a** the liver and **b** the lung, normalized to a range of 0–1 in each 1 Mb bin. The horizontal bar plots display the number of EVCs in each bin (bins with no EVCs appear black in the heatmap). The plots in the bottom panel display the chromosome-wide means of cytosine modification as a function of ZT, fitted using the harmonic regression model. The shading around the regression lines represents the 95% confidence band. All data were double plotted to aid with visualization. ZT, Zeitgeber time; modC, cytosine modification

determination (i.e., $R^2$ of the harmonic regression fit) of all EVCs within that gene. Our observations were threefold.

First, mRNA levels positively correlated with their corresponding cytosine modification oscillations in liver and lung tissues (weighted Pearson's $r = 0.14$ and $0.19$; $p = 4.1 \times 10^{-6}$ and $1.4 \times 10^{-10}$, respectively), suggesting that genes with more robust oscillations tend to be more abundantly expressed (Supplementary Data 12-13). Second, circadian oscillations of mRNA were positively

correlated with oscillations of their corresponding gene modifications (weighted Pearson's $r = 0.075$ and $0.19$; $p = 0.015$ and $4.3 \times 10^{-10}$, for the liver and lung, respectively), indicating that circadian transcripts tend to oscillate together with osc-modCs in their gene body. Third, acrophases of mRNAs were shifted by 0-6 h (range of significant phase shifts with Bonferroni corrected permuted $p < 0.05$) from the nadir (ZT when oscillation reaches its minimum) of their matched, modC densities (Fig. 5a-b and

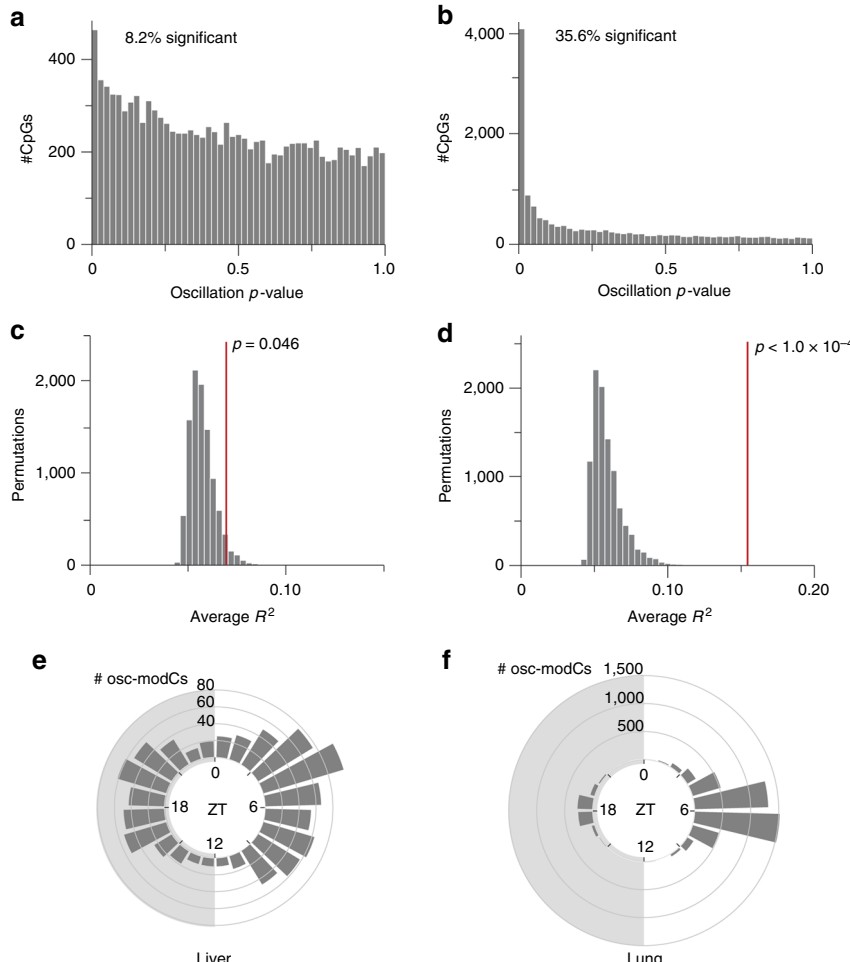

**Fig. 4** Characterization of osc-modCs in the 9-mo mouse liver and lung. **a**, **b** The $p$-value histogram of harmonic regression fits on individual cytosines showing that 8.2% (983 of 11,941) and 35.6% (5,054 of 14,199) of EVCs were oscillating ($p < 0.05$) in **a** the liver and **b** the lung, respectively. **c**, **d** Average proportion of variance explained ($R^2$) by the harmonic regression fits across all EVCs in each of 10,000 permutations of ZT labels. The red line depicts the observed average $R^2$ in **c** the liver and **d** the lung. **e**, **f** Acrophase rose plot showing modification peak times of osc-modCs in **e** the liver and in **f** the lung. osc-modC, oscillating cytosine modifications

Supplementary Fig. 7), suggesting a temporal relationship between the two events. As circadian mRNAs only partially reflect oscillations in nascent transcription[11] and given the regulatory proximity of cytosine modification to nascent transcription, osc-modCs may be more robustly associated with oscillations in nascent transcription than mature mRNA.

Sequences flanking osc-modCs contained both canonical (CANNTG) and non-canonical (CANNNTG, GANNTG) E-box motifs[21] ($e$-value $= 8.3 \times 10^{-26}$-$5.2 \times 10^{-212}$; Fig. 5c-d and Supplementary Data 14-15). E-box response elements play key roles in regulation of circadian transcripts[5,6]. It has been shown that Myc-MAX heterodimer complex can interact with E-box motifs[22] and Myc can in turn interact with DNA methyltransferase 3A[23] to methylate proximal CpG sites. In all, our findings show that osc-modCs are intricately linked to circadian transcriptomics.

**Differential acrophase timing of osc-modCs**. We categorized osc-modCs into "sleep acrophases" (ZT0-12) and "wake acrophases" (ZT12-24) (Fig. 3i-j), and observed that the acrophase time was associated with the osc-modC average modification density. In both liver and lung tissues, osc-modCs with wake acrophases showed a significantly higher average cytosine modification density (mean ± SE; 67 ± 1.20% and 62 ± 0.74%, respectively) compared with osc-modCs with sleep acrophases

(54 ± 1.25% and 50 ± 0.36%, respectively). Modification densities of oscillating cytosines from the two acrophase peaks exhibited the largest modification differences during ZT12-24, whereas during ZT0-12 the densities were closest to each other (Fig. 5e-f). Borrowing from the field of astronomy, the cytosine modification densities were at their "apogee" at ~ ZT18, as the distance between the two sinusoidal functions arrived at its maximum, and conversely reached their "perigee" at ~ ZT6, as the distance between the two arrived at its minimum.

**Osc-modCs' association with age-correlated changes**. Although aging and the disruption of circadian processes are closely linked[3,4], the molecular mechanisms of this association are not clear. It is conceivable that DNA modification can provide the platform to mediate this association due to its role in both the aging and circadian processes. Therefore, we investigated whether osc-modCs are related to age-correlated cytosine modifications (age-modCs).

We used a linear regression model to detect cytosine modification changes that either increased or decreased with age in the 9-, 15-, and 25-mo mice. We found that liver samples had more age-modCs compared with the lung (24.1% and 8.4% of EVCs, with $p < 0.05$ after Bonferroni correction, in the liver and lung, respectively; Supplementary Fig. 8 and Supplementary

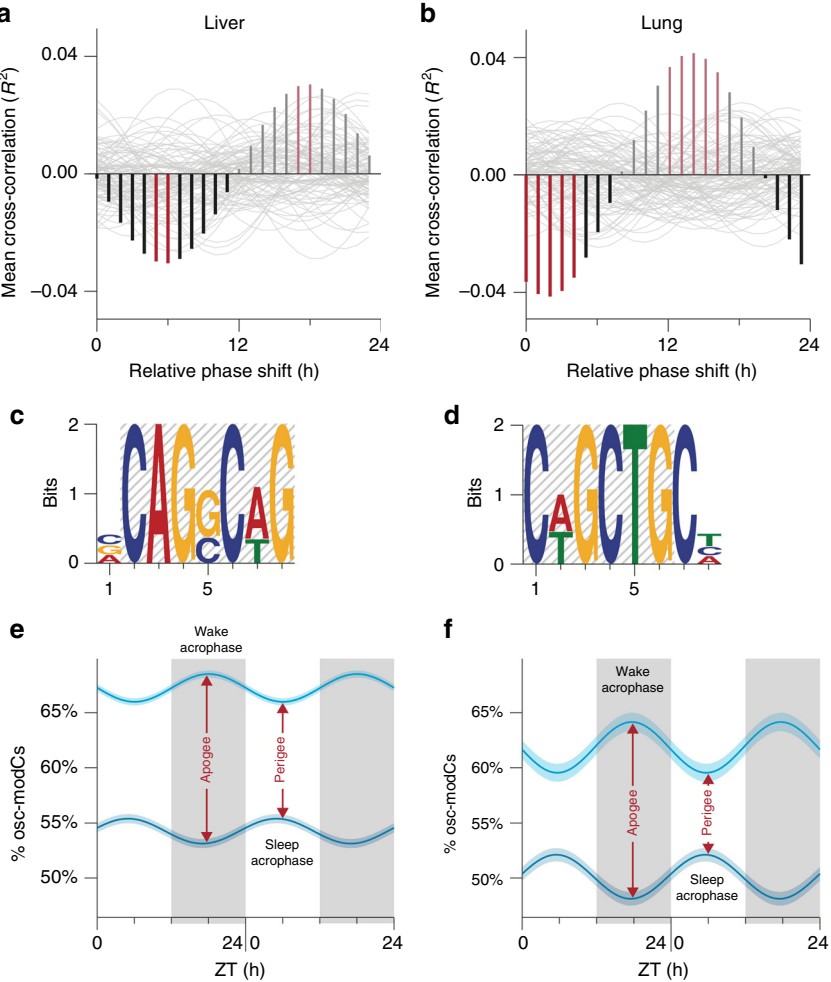

**Fig. 5** The transcriptional and temporal coordination of oscillating modified cytosines in the 9-mo mouse liver and lung. **a**, **b** Mean gene-body modification densities of epigenetically variable cytosines in **a** the liver and **b** the lung were cross-correlated with corresponding circadian mRNA profiles after each iteration of a forward 1 h phase shift in the mRNA profiles. Bars represent the observed mean cross-correlation between mRNAs and modification densities for a given phase shift (red bars indicate Bonferroni corrected permuted $p < 0.05$). The positive faded bars are mirror images of the negative bars (by nature of correlating phase shifted waves, a wave shifted by half a period anti-correlates with itself). Fine gray curves in the background depict cross-correlations of 100 randomly selected (out of 10,000) permutations where both ZT labels and mRNA-gene modification density pairs were shuffled. **c**, **d** Representative E-box motifs enriched within 100 bp in either direction of osc-modCs in **c** the mouse liver and **d** the lung. **e**, **f** Harmonic regression fits on the average modification density of osc-modCs with wake and sleep acrophases in **e** the mouse liver and **f** the lung, as a function of ZT. Shading around the regression lines represents the 95% confidence band. ZT, Zeitgeber time; modC, cytosine modification

Data [16-17]). Osc-modCs from the 9-mo mice showed a strong association with age-modCs in both the liver and lung tissues (OR = 2.3 (2.0-2.7) and 1.4 (1.2-1.6); $p = 2.6 \times 10^{-24}$ and $4.0 \times 10^{-5}$, respectively). In addition, the circadian amplitudes were correlated with the magnitude of the epigenetic aging effects and accounted for 18.4% (Pearson's $r = 0.43$; $p = 1.3 \times 10^{-14}$) and 72.8% (Pearson's $r = 0.85$; $p = 3.4 \times 10^{-111}$) of the age-dependent variance, in the liver and lung, respectively (Fig. 6a-b).

The associations between sleep or wake acrophases and age-dependent gain or loss of cytosine modification were highly asymmetric; cytosines with sleep acrophases that exhibited an aging effect showed increased modification with age, whereas those with wake acrophases predominantly lost modification with age (OR = 68 (30-166) and 394 (113-1,875); $p = 1.4 \times 10^{-44}$ and $9.6 \times 10^{-46}$, in liver and lung tissues, respectively; Fig. 6c-f). The association between osc-modC acrophase time and direction of change in age-modC may provide insights into why genomic elements that are polarized in terms of DNA modification densities converge toward the mean in the aging epigenome[24,25].

If osc-modCs have a causative influence on age-dependent epigenetic changes, the former would precede the latter chronologically. Therefore, osc-modCs that are exclusively present in younger animals (i.e., 9-mo) should be enriched for cytosines whose modifications show an aging trend after 9 months. Conversely, if aging induces osc-modCs, age-modCs should become more abundant amongst the osc-modCs specific to the older groups (i.e., 15- and 25-mo). Consistent with a circadian causative direction, cytosines oscillating only in the 9-mo showed an enrichment of age-modCs (binomial $p = 3.5 \times 10^{-8}$; 12.4% age-modCs), whereas cytosines oscillating only in the 15-mo or only in the 25-mo showed no enrichment or even depletion of age-modCs (binomial $p = 0.42$ and 0.0018; 7.6% and 4.3% age-modCs, respectively; Fig. 6g). Interestingly, osc-modCs common to all three age groups also showed a significant depletion of age-modCs (binomial $p = 0.0057$; 5.3% age-modCs), suggesting that the proposed circadian-aging conversion has not occurred yet but may take place in the animals living longer than 25 months. We repeated this analysis using matched sample sizes or matched proportions of osc-modCs across all age groups and

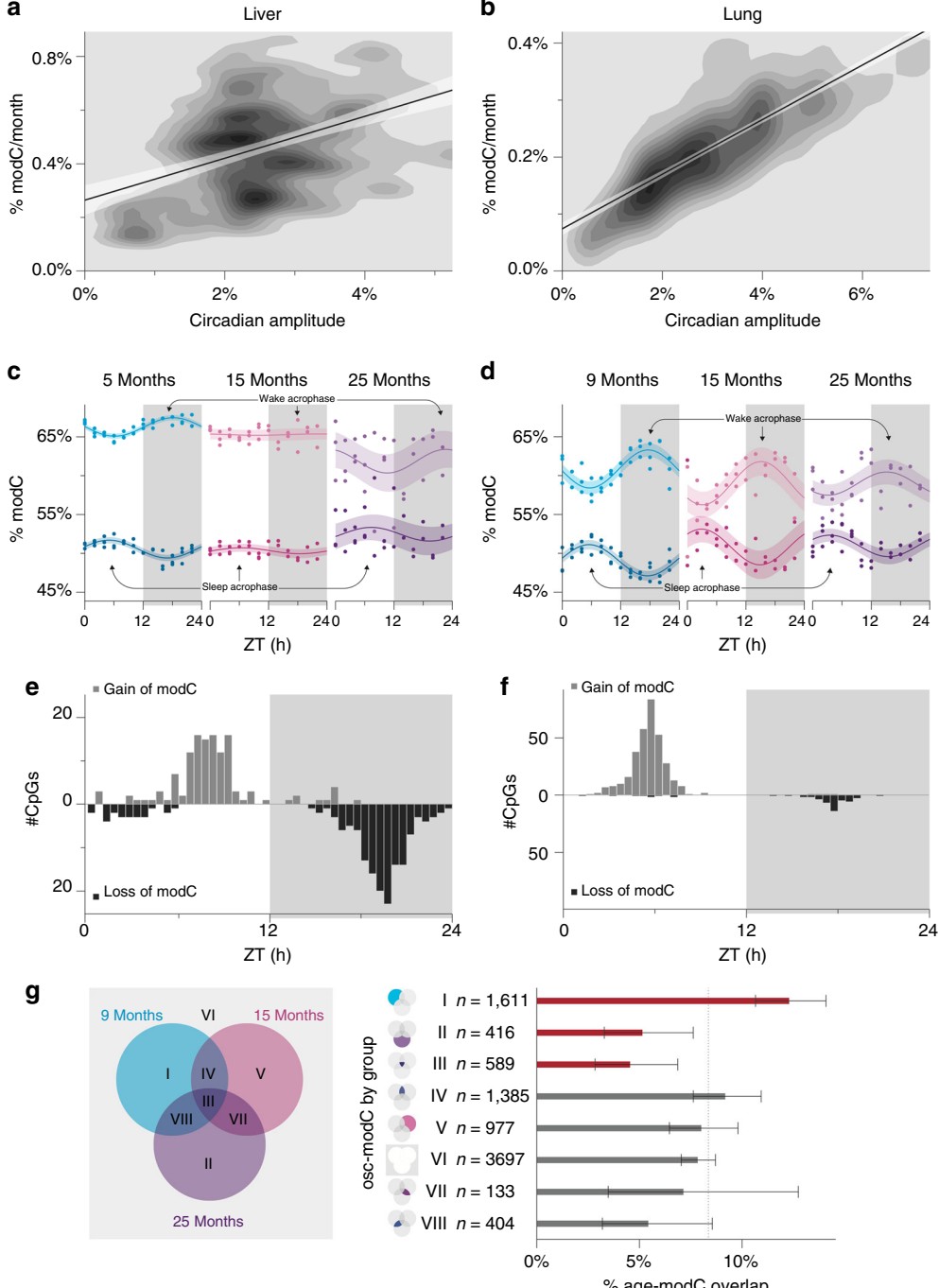

**Fig. 6** The relationship between oscillating and age-correlated cytosine modifications in the mouse liver and lung. **a, b** Density contour plots of the age-modC magnitude and the osc-modC amplitude in **a** the liver and **b** the lung. Contours were rendered using 2D kernel density estimation and the lines represent linear regression fits. Shading around the regression lines represents the 95% confidence band. **c, d** The mean modification levels of 9-mo osc-modCs, stratified by wake and sleep acrophases, converge with increasing age in the **c** the liver and **d** the lung. The curves show harmonic regression fits of the mean cytosine modification and the shading around the regression lines show 95% confidence bands. **e, f** The relationship between acrophase time and aging direction in **e** the liver and **f** the lung. The white areas represent periods of lights on (i.e., sleep), whereas the shaded areas show periods of lights off (i.e., wake). The gray bars show counts of cytosines with gain of modification with age, whereas the black bars show counts for loss of modification. **g** Osc-modC were categorized according to their placement in the Venn diagram. The bar plots represent the percentage of cytosines in each group that were also age-modCs with 95% confidence intervals. The dashed line represents the expected proportion of age-modC and the red bars show significant (*p* < 0.05) deviation from this expected line. Age-modC, age-correlated cytosine modifications; Osc-modC, oscillating modified cytosines; modC, modified cytosines; ZT, Zeitgeber time

arrived at the same conclusions (Supplementary Fig. 9). These findings suggest that oscillations of cytosine modification could precede age-correlated cytosine modification changes, but additional experiments are required to establish causality.

As epigenetic factors are intimately related to transcription, we asked whether associations between circadian rhythmicity and aging also extend to the transcriptome. We compared publicly available circadian[7] ($n = 72$ for liver and lung) and aging[26,27] ($n = 7$ and 9 for liver and lung, respectively) transcriptomes (Supplementary Data 18-19) of mouse tissues and detected a significant overlap between the two (OR = 1.3 (1.2-1.4) and 1.6 (1.5-1.8); $p = 1.3 \times 10^{-8}$ and $1.3 \times 10^{-25}$ for the liver and lung, respectively). We also found that amplitudes of oscillating transcripts correlate with magnitudes of aging effects ($r = 0.65$ and 0.56; $p = 4.4 \times 10^{-170}$ and $9.4 \times 10^{-83}$ in the liver and lung, respectively) (Supplementary Fig. 10a-b). Gene Ontology (GO) analysis of the mRNAs common to both data sets showed enrichment of various terms including catabolic and metabolic processes in the liver (false discovery rate (FDR) $q = 0.05$-$8.9 \times 10^{-24}$; Supplementary Fig. 10c and Supplementary Data 20), and cell adhesion and migration in the lung (FDR $q = 0.05$-$1.5 \times 10^{-6}$; Supplementary Fig. 10d and Supplementary Data 21). These findings suggest that the circadian-aging association could be universal and therefore also detectable in metabolomic and proteomic studies.

## Discussion

The evidence for daily cyclical patterns of cytosine modifications reported in this study makes cytosine modification a bona fide member of the cellular circadian machinery alongside oscillating RNAs, histone modifications, proteins, and metabolites. The underpinnings of the circadian regulation include numerous interdependent feedback loops, which precludes a clear demarcation of the hierarchy of such elements. Consideration of the temporal dimension is necessary for unravelling the complex molecular circuitries of the cell.

Circadian patterns of cytosine modification may help elucidate several poorly understood epigenetic phenomena. Osc-modCs can, in part, explain the occurrence of ongoing active demethylation and production of hmC in differentiated somatic cells.

Osc-modCs can explain a large fraction of variation in cytosine modifications and therefore this work indicates that circadian mismatched samples could result in findings that are confounded by oscillations. Failure to perform experiments in a circadian-informed manner may conflate epigenetic (alongside transcriptomic, metabolomic, and proteomic) differences arising from circadian mismatches with those attributed to underlying traits.

Epigenetic variation due to cytosine oscillation may even be larger than our estimates. In our study, we captured the composite effect of oscillations that occur in a large population of cells. Individual cells may exhibit differences with respect to the specific cytosine positions that undergo circadian oscillations within a genomic region. These subtle intercellular variations in daily reprogramming can create epigenetic heterogeneity within a specific region or genomic element. This could be a potential mechanism generating intermediate cytosine modifications in regulatory elements such as DNase hypersensitive sites and enhancers[28].

We observed similarities and distinct differences in aging and osc-modC profiles between the mouse liver and lung tissues, namely a smaller number of osc-modCs but a larger number of age-modCs in the liver. It is possible that a faster rate of aging in liver can lead to diminished oscillations and this difference in signal-to-noise ratio would reduce the robustness of the osc-modC/age-modC association in the liver.

At present, opinions regarding the contribution of circadian dysfunction to aging differ significantly[3,29]. We detected that osc-mCs are strongly associated with age-dependent changes in the epigenome and attempted to uncover the direction of such association. To address the relationship between the two epigenomic phenomena, Bradford Hill's causality criteria[30] may be used. Four of the criteria supported a causal role for osc-modCs in aging by a varying extent: (I) circadian oscillation of cytosine modifications preceded age-modC (temporality); (II) amplitudes of circadian oscillation correlated with the size of the aging effect (gradient); (III) both cytosine modification and transcriptomes showed robust circadian-aging associations (consistency); and (IV) it is conceivable that daily epigenetic reprogramming errors gradually accrue over time converting the circadian epigenome into the aging epigenome (plausibility).

The circadian DNA modification may shed a new light on the associations between aging epigenomes and complex diseases, especially the ones with late age of onset. Deterministic, and not stochastic, accumulation or loss of modCs during the lifetime of an individual explains why DNA modification markers can be precise predictors of biological age[31]. Accelerated epigenetic aging, which is associated with increased morbidity and mortality[32–34], may be a result of deviated trajectories of circadian DNA modification. Our finding of the circadian origin of epigenetic convergence is relevant not only to aging but also to carcinogenesis; it suggests a common mechanism between age-dependent DNA modification changes and cancer epigenome[35]. Hypothetically, even a small (e.g., 0.1%) unidirectional daily asymmetry in circadian DNA demethylation–remethylation cycle can result in a bona fide epigenomic state of a malignancy in several months.

The circadian clock evolved before the advent of air travel, light bulbs, and constant food availability; it does not contain the necessary counterbalances for these stressors[36]. The plasticity and programmability of cytosine modification and the circadian clock make them amenable to continuous environmental inputs that can ultimately facilitate both circadian dysregulation or the correction of such dysregulations. Circadian parameters can be altered by various factors, including diet and chemical compounds, and such circadian manipulations may have an impact on the molecular trajectories during aging. In other words, it may be possible to potentially influence aging outcomes by modifying circadian epigenomes and transcriptomes at a younger age.

## Methods

**Sample information.** C57BL/6JRj male mice were obtained from Janvier Labs. The animals were singly housed in polypropylene cages ($44 \times 22 \times 19$ cm) to facilitate assessment of activity. They were given ad libitum access to food, water, and a 17 cm diameter running wheel mounted in the cage. The animals were entrained to a 24 h light–dark cycle maintained at 12 h light and 12 h dark, where ZT0 refers to the light onset, for a minimum of 30 days. Wheel running activity was monitored using Vitalview (Phillips-Respironics) and circadian entrainment (i.e., wheel running activity onset) was verified using Actiview (Phillips-Respironics). At ages 9, 15, or 25 months (mo), mice were killed by cervical dislocation every 2 h over the course of 58 h (minimum). Collected tissues were snap frozen in liquid nitrogen and stored at $-80$ °C.

**Ethical approval.** All experiments were approved by the Centre for Addiction and Mental Health Research Ethics Board (protocol 567), the University of Toronto Animal Care Committee (protocol 20010315), and performed in accordance with relevant guidelines and regulations.

**DNA and RNA extraction.** Frozen mouse tissues were digested in lysis buffer (35 mM EDTA, 75 mM NaCl, 10 mM Tris-HCl pH 8.0, 1% SDS, and 2 mg ml$^{-1}$ Proteinase K) overnight and extracted using the phenol : chloroform method. Genomic DNA quality and quantity were examined on a 1% agarose gel, a NanoDrop 2000 Spectrophotometer (Thermo Fisher Scientific), and a Qubit 2.0 fluorometer (Invitrogen).

For RNA studies, frozen tissues were treated with RNAlater-ICE Frozen Tissue Transition Solution (Ambion) and total RNA was extracted with an RNeasy Mini Kit (Qiagen). Total RNA was quantified using a NanoDrop 2000 Spectrophotometer (Thermo Fisher Scientific) and an Agilent 2100 Bioanalyzer with a RNA 6000 Nano Kit (Agilent). The investigated RNA samples had minimum RNA Integrity Number of 7.5.

**Bisulfite and oxBS conversion**. A total of 750 ng of genomic DNA was bisulfite converted using an EZ DNA Methylation Kit (Zymo) according to the manufacturer's protocol for the HumanMethylation450 BeadChip (Illumina), with the following modifications suggested by the manufacturer for a more stringent conversion: 7.5 µl of M-dilution buffer was used for the reaction, which was incubated at 42 °C for 30 min before addition of the CT-Conversion Reagent. A total of 185 µL of the M-dilution buffer was used in the preparation of the CT-Conversion Reagent, and only 97.5 µlL of the reagent was added per reaction. oxBS conversion was performed using a TrueMethyl kit (CEGX) as per the manufacturer's recommendations. Using CpH methylation as a proxy for bisulfite conversion efficiency, the mean non-conversion rate for all experiments were estimated at 0.6 ± 0.6% (mean ± SD).

**Bisulfite padlock probe and library preparation**. Bisulfite padlock probes were designed using the ppDesigner 2.0 software[16]. The reference mouse genome (mm10) was masked for genomic variations and repeats (dbSNP 138, Microsatellites, RepeatMasker, Segmental Dups, Simple Repeats, and WindowMasker + SDust). For the remaining (non-repetitive) chr 7 sequence, all possible probe arms were designed and filtered for the presence of at least one CpG and less than 10% of the masked genomic sequence within the targeted region. Seven basepair unique molecular identifiers (UMIs) were added to the common sequence of each probe immediately adjacent to the probe annealing arms and later used for removal of PCR duplicates. The probes were printed by CustomArray (Bothell, WA, USA) and prepared according to the published protocol[16] with minor modifications.

Briefly, 1–100 nM of the synthesized probes were amplified using 400 nM of pAP1V61U (5′-G*G*G TCATATCGGTCACTGTU-3′) and AP2V6 (5′-5Phos-CACGGGTAGTGTGTATCCTG-3′) primers, and 1 × KAPA SYBR Fast qPCR mix in four separate 50 µl reactions. The following cycling conditions were used: 95 °C for 30 s, 15 cycles of 95 °C for 10 s, 55 °C for 20 s, and 70 °C for 30 s, with a final extension of 70 °C for 2 min. The amplicons were pooled and purified using QIAquick PCR purification kit (Qiagen) following the manufacturer's recommendation. The final product (0.2 nM) was used as template for a large-scale production PCR involving a minimum of four 96-well plate reactions amplified in identical conditions to the first round. The amplicons were then pooled and concentrated using ethanol precipitation. The amplicons were re-purified using QIAquick PCR purification kit (Qiagen) following the manufacturer's recommendation. Amplification adaptors were then removed using three enzymatic digestions. First, 15–20 µg of the purified amplicon were mixed with 50 units of lambda exonuclease (New England Biolabs) in a 150 µl reaction containing 1 × lambda exonuclease buffer and incubated for 1 h at 37 °C, to remove the bottom strand. The digested amplicons were purified using ssDNA/RNA clean & concentrator kit (Zymo) using the manufacturer's protocol. Second, 3–5 µg of the single-stranded probes were then digested with 5 units of USER (New England Biolabs) in an 80 µl reaction containing 1 × DpnII buffer (New England Biolabs) and incubated for 1 h at 37 °C. Next, 5 µl of 100 µM guide oligo (5′-GTGTATCCTGATC-3′), 2 µl of 10 × DpnII buffer and 8 µl of water were added to the mix and the reaction was incubated at 94 °C for 2 min, followed by 3 min at 37 °C. Third, 250 units of DpnII (New England Biolabs) were added and the reaction was incubated at 37 °C for 2 h followed by heat inactivation at 75 °C for 20 min. The probes were then purified using a TBE-Urea denaturing gel and cutting the band corresponding to ~ 120 bp.

For the padlock library preparation, 1.5 ng of the purified probes was mixed with 200 ng bisulfite-treated genomic DNA (quantified using Qubit ssDNA Assay Kit) in a 20 µl reaction containing 1 × ampligase buffer and covered with 20 µl of mineral oil to prevent evaporation. The reaction was then incubated at 94 °C for 30 s and gradually lowered (− 0.5 °C/25 s) to 55 °C and incubated for an additional 20 h. With the plate still in the thermocycler, a 6.5 µl mixture containing 2.5 µl 10 × NAD⁺ (NEB), 0.85 µl dNTP (1 mM), 0.85 µl ampligase (Epicentre, Madison, Wisconsin, USA), 0.85 µl 10 × ampligase buffer, and 1.5 µl of preheated PfuTurbo Cx Hotstart DNA Polymerase (Agilent Genomics) was added. The unligated products were removed using 20 units of exonuclease I and 200 units of exonuclease III (Epicentre). The circularized DNA was then enriched by PCR using 5 µl of the reaction in a 50 µl volume containing 200 nM of Amp_F_SE and SE_Amp_IndX indexing primers (Supplementary Data 22) and 1 × KAPA SYBR Fast qPCR master mix using a StepOnePlus Real-Time PCR System (Thermo Fisher Scientific) at the following cycling conditions: 95 °C (30 s); 8 cycles of 95 °C (10 s), 58 °C (30 s), and 72 °C (20 s); 15 cycles of 95 °C (10 s) and 72 °C (20 s); and a final extension at 72 °C (3 min). The PCR products were purified using 0.7× volume of AMPure magnetic beads (Beckman Coulter) with two 70% ethanol washes, and quantified using a Qubit dsDNA HS assay (Thermo Fisher Scientific). Equal amounts of each sample were pooled, and the band at ~ 360 bp was excised and purified using standard agarose gel extraction methods. The purified libraries were quantified for sequencing using KAPA Library Quantification kits.

**Preprocessing of sequencing data**. The libraries were sequenced on a HiSeq 2500 platform (Illumina) at 2 × 125 paired-end reads by using custom sequencing primers (Supplementary Data 22). For each FASTQ file, the UMIs were removed from the start of the reads and saved for later processing. The FASTQ reads were quality trimmed using Trimmomatic[37] for trailing bases with a phred score < 30 and all reads with post-trimming length < 50 bp. The trimmed reads were then aligned to a masked genome (mapping only to regions within a 100 bp window of the known probe locations) using Bismark v0.14.3[38] and Bowtie 2 v2.2.2[39]. The aligned reads were further filtered to only include reads that had start and end positions matching the padlock probe annealing arm sequence with no more than one mismatch. The filtered reads were subsequently PCR de-duplicated using the UMIs. The filtered read pairs were then converted to modification calls using the Bismark methylation extractor tool[38].

Samples with low coverage were removed based on a comparison relative to other samples within the same experiment. Individual CpGs were required to have a minimum coverage of 30 reads in each sample for inclusion in the analysis. β-Values were calculated as the proportion of cytosines and thymines for a given CpG; $\beta = M/(M + U)$, where M = number of cytosines (modCs) and U = number of thymines (unmodified cytosines). All samples for a given age and tissue were internally correlated to identify outliers, and samples with average inter-sample correlation value more than 3 SD below the mean were excluded from further analysis.

hmC data were derived from the set of cytosines which were epigenetically variable (see "Detection of EVC in mouse tissues") in either mC (from oxBS padlock sequencing) or modC (from bisulfite padlock sequencing) data sets. hmC values were estimated by subtracting the oxBS sequencing densities from the bisulfite sequencing densities following the TrueMethyl kit (Cambridge Epigenetix, Cambridge, UK) recommendations.

Lung mC data were obtained from a larger set of probes that, in addition to chromosome 7, targeted other chromosomes. Only probes overlapping with the primary chromosome 7 probe list were used for analysis. Outlier samples were calculated and removed prior to subsetting.

**Detection of EVCs**. To detect EVCs, whose biological signal exceeded technical variation, we analyzed three technical replicates for each biological sample. A one-way analysis of variance test comparing the variance between technical and biological replicates was performed on every cytosine. Each tissue and age groups were analyzed separately and cytosines with $p < 0.05$ were identified as EVCs. Following EVC identification, the cytosine modification values of the three technical replicates were averaged using the median, in order to obtain a single robust biological data point for further analyses.

**Detection of circadian oscillations**. A harmonic linear regression model was used to identify circadian oscillations. The period was fixed to 24 h, and the phase and the amplitude were modeled as a linear combination of sine and cosine terms as follows:

$$y = b_0 + b_1 \sin\left(\frac{2\pi ZT}{24}\right) + b_2 \cos\left(\frac{2\pi ZT}{24}\right) + \epsilon. \tag{1}$$

where $y$ is the observed modification level, $b_{0-2}$ are regression coefficients, ZT is the time of observation, and $\epsilon$ is the error term. P-values were obtained by comparing this model to the null model without the sine and cosine terms using an F-test. Cytosines with harmonic regression $p < 0.05$ were identified as osc-modCs with their amplitude ($A$) and acrophase ($\phi$) defined as:

$$A = 2\sqrt{b_1^2 + b_2^2}, \tag{2}$$

$$\phi = \frac{12}{\pi} \text{atan2}(b_1, b_2) \bmod 24. \tag{3}$$

To determine whether the average proportion of variance explained by oscillations is higher than expected by chance, 10,000 permutations were performed by shuffling ZT labels and the mean $R^2$ value was calculated across the EVCs in each permutation. The permutation $p$-value was derived as a fraction of permutations with the permuted mean $R^2$ value greater than the observed.

PCA was used to quantify the amount of variability explained by oscillations. Principal components were calculated via singular value decomposition of the mean centered data matrix. The resulting scores of four main principal components were inspected for oscillations by fitting the harmonic regression model described above.

**Analysis of circadian and tissue-specific transcripts**. mRNAs of two key circadian genes, period circadian clock 2 (Per2) and aryl hydrocarbon receptor nuclear translocator like (Arntl), were used to confirm the circadian entrainment of mice[1]. Liver-, lung-, and macrophage-specific mRNA targets were selected using a public microarray expression data set (Gene Expression Omnibus (GEO) accession:

GSE1133[40]). The data were queried for genes whose expression exhibited the highest fold change between the tissue of interest and other tissues. Absence of circadian variation in the mRNA transcripts specific to hepatocytes, pneumocytes, and macrophages was verified using the CircaDB database[41].

Quantification of relative mRNA levels was performed on an Applied Biosystems ViiA 7 Real-Time PCR System (Applied Biosystems). First-strand complementary DNA synthesis was performed with a SuperScript III First-Strand Synthesis System (Thermo Fisher Scientific) on DNase I-treated RNA. Real-time PCR was performed in triplicate using Power SYBR Green PCR Master Mix (Applied Biosystems), with 400 nM primers (Supplementary Table 3) in a total reaction volume of 10 µl and at the thermal cycling conditions recommended by the manufacturer. Glyceraldehyde-3-phosphate dehydrogenase (Gapdh) mRNA levels were used as endogenous controls. The results were analyzed using ViiA 7 software and R[42].

**Analysis of the transcriptomic datasets**. Pre-normalized public circadian transcriptomic microarray (GEO accession: GSE54650[7]) and aging transcriptomic data sets (GEO accession: GSE57809[26] and GSE6591[27] for the liver and lung, respectively) were used for detection of circadian (see "Detection and analysis of circadian epigenetic oscillations") and nominally significant aging mRNAs (see "Aging analysis for the mouse tissue DNA samples"). Data sets were matched by RefSeq ID or gene symbol and genes with missing mRNA data were removed.

GO enrichment analysis on the public transcriptomic dataset was performed using GREAT 3.0[43]. A hypergeometric test was performed using oscillating and aging transcripts as foreground and all transcripts as background. The resulting p-values were FDR adjusted and significance threshold was set as $q < 0.05$. To reduce the dimensionality of the GO enrichment terms to the most informative ones, the list of enriched terms ($q < 0.05$) was submitted to REViGO[44] and terms with similar gene lists were reduced by merging terms with dispensability score < 0.7.

Pearson's correlations between circadian transcriptomic data and median $R^2$ of cytosine modification data were weighted by the number of cytosines within the gene.

**Cytosine modification and circadian mRNA phase shift**. Each mRNA transcript was paired with the corresponding EVCs within its gene body. For each transcript, the harmonic model estimate from mRNA was correlated with the mean cytosine modification values within the same gene (Pearson's correlation). The overall correlation was summarized as the mean correlation value across all mRNA transcripts. In order to estimate the strength of each phase shift between mRNA and modification, the above procedure was repeated 24 times, each time shifting the obtained harmonic model fit by 1 h. The procedure for a single phase shift can be represented as:

$$y_{ip} = b_{0i} + b_{1i}\sin\left(\frac{(ZT - p)2\pi}{24}\right) + b_{2i}\cos\left(\frac{(ZT - p)2\pi}{24}\right) + \epsilon_i, \quad (4)$$

$$r_{ip} = \text{cor}(x_i, y_{ip}), \quad (5)$$

$$r_p = \frac{\sum_{i=1}^{N} r_{ip}}{N}. \quad (6)$$

where $y_{ip}$ is a vector of mRNA estimates at phase shift $p$, for each time of observed cytosine modification, ZT, for gene $i$. $b_{0i-2i}$ are the mRNA harmonic regression coefficients and $\epsilon_i$ is the error term. cor is the function for Pearson's correlation coefficient and $x_i$ is a vector of mean modification values. The summarized strength of the correlation for phase shift $p$ is estimated by $r_p$, the mean of all genes' Pearson's r, $r_{ip}$.

To generate a null distribution, permutations ($N = 10,000$) were performed by shuffling the ZT times and pairing of mRNA and modification values. Each permutation produced 24 overall correlation estimates (i.e., 10,000 values for each phase shift) and for each phase shift the permutation p-value was calculated as the fraction of permutations with overall correlation greater than the observed value. The permutation p-values were corrected for multiple testing using the Bonferroni procedure.

**Aging analysis**. Only cytosines that were epigenetically variable across all three age groups were considered for aging effects. Cytosines exhibiting age-dependent modification across the three age groups (9-, 15-, and 25-mo) were identified using an F-test between a null intercept-only model ($y_{null}$) and a linear model using age as a predictor ($y_{alternative}$) defined as:

$$y_{null} = b_0 + \varepsilon, \quad (7)$$

$$y_{alternative} = b_0 + b_1\text{age} + \varepsilon. \quad (8)$$

Cytosines whose modification showed a significant correlation with age (Bonferroni corrected $p < 0.05$) were called age-correlated cytosines (age-modC)

and the slope of the regression line (coefficient $b_1$ in $y_{alternative}$ model) was used to determine the direction of change.

**Motif analysis**. Sequence motifs were examined at the oscillating cytosine position ± 100 bp. Overlapping 200 bp regions (i.e. redundant sequences) were merged into one sequence. MEME suite 4.10.2[45] was used to identify overrepresented sequences using the following parameters: -dna, -mod anr, -maxsites 1000, -nmotifs 20, -evt 1e-10, -revcomp, -maxsize 1,00,00,000.

**Analysis of differences in acrophase timing**. Absolute acrophase differences (minor arc length) between paired data sets (liver 9-mo modC and lung 9-mo modC; liver hmC and mC; and lung hmC and mC) were calculated for each cytosine and averaged by taking the median. Permuted values were calculated by randomly shuffling acrophase pairings of one data set relative to the other and again computing median absolute acrophase differences. P-value was measured as the number of permutations with a value greater than (or less than, depending on the alternative hypothesis) observed in the real data.

**Circadian-aging association analysis**. Associations between osc-modC and aging probes were estimated using two-sided Fisher's exact test. Only EVCs were used to compute the contingency table. All computational analyses were performed in R[42] unless specified otherwise.

**Data availability**. Next-generation sequencing data that support the findings of this study have been deposited in the GEO with the accession code GSE83947.

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

## Acknowledgements

We thank E. Ellison for consultation regarding blood cells in solid tissues, T. Khare for advice on bisulfite padlock sequencing experiments, A. Lim for advice on circadian analyses, A. Wong for help with animal studies, and A. Krisciunas and P. Gibas for discussions regarding data analysis. We thank I. Meirelles, A. Norwood, and J. Yao for their consultations and assistance in generating the figures. This work was supported in part by the Canadian Institutes for Health Research, the National Institute of Mental Health, Brain Canada, and the Krembil Foundation (A.P.). J.G. and K.K. were funded by a grant (MIP-043/2014) from the Research Council of Lithuania.

## Author contributions

A.P. conceived the project. G.O., S.E., and A.P. developed the theoretical framework and designed the experiments. S.E., A.Z., A.N., G.O., V.L., P.J., E.S.O., R.H.J., and M.S. performed the wet lab experiments. T.C.S. and M.R.R. performed the entrainment and monitoring of mice. G.O., M.C., J.G., K.K., and A.P. were responsible for the data analysis, presentation, and interpretation of the results. D.E.G. and G.O. developed the algorithms to process the sequencing data. G.O. and A.P. wrote the manuscript, with input from all co-authors.

## Additional information

**Competing interests:** The authors declare no competing financial interests.

