## [Peer Review File · Nature Communications]

Reviewers' comments:

Reviewer #1 (Remarks to the Author):

The paper describes a series of meticulous experiments investigating the diurnal variation of cytosine variation in the mouse liver and lung.

The authors entrain the circadian cycle in mice, extract liver and lung nucleic acids, and concentrate the cytosine content of chromosome 7. They show that 50% of Cytosines vary and using PCAs show that the lung and liver have different patterns of variation. They further show that transcription variation partially follows cytosine modification, with a difference in lag time between tissues.

They go on to examine the cytosines that vary with age, and show an age-related diminution of cytosine cycling and a partial overlap between these Age-modC and Osc-modC.

The study of isolated neutrophils from a 52 year old man introduces a further level of complexity, and a range of technical questions about the likely effects of neutrophil isolation from other white cells on modC and transcription.

Analytically, there is much to be admired in the paper, but the cost is that it makes it very demanding for the general reader. Harmonic regressions, double plotted data and glide reflections are examples of concepts that need explaining.

The overlap between oscillation and age is not surprising, given that 50% of cytosines vary diurnally, and a statistical proof that this is not by chance is essential. The CpGs that vary with age in the human genome are well documented (for example by Horvarth) and it would be good to know if these coincide with the murine loci.

I am not sure about the value of including the human leukocyte data. It fits uneasily with the liver and lung focus of the murine work and, although interesting, it raises a fresh set of questions rather than consolidating the main study.

Reviewer #2 (Remarks to the Author):

Circadian oscillation of epigenetic cytosine modifications

In this manuscript the authors reveal an interesting and novel role of cytosine modifications under circadian control. These modifications have been previously thought to be mostly stochastic. However, recent epigenetic papers have shown that for example gene expression is controlled and regulated by these modifications where they explain up to 50% of all variation in gene expression and the stochastic hypothesis of epigenetic mechanisms does not explain everything. I support the publication of this manuscript as it has interesting novel findings, but suggest a revision for this manuscript with the following edits.

1. The authors should emphasize the biological relevance of cytosine modifications also as regulators for gene expression and examine whether the osc-modCs defined in this paper are localized on known transcription factor binding sites. The authors even state on line 203 that "Methylation of E-box response elements can inhibit transcription factor binding²⁴ 204 , which suggests that osc-modCs may be involved in regulating the binding of 205 transcription factors to E-box elements. In all, our findings show that osc-modCs are intricately 206 linked to circadian transcriptomics." Formal analysis should be done to give an answer to this question as it is one of the key outcomes of the analysis.

2. N for the public data sets should be given in the text and figures to give the reader an understanding of the power to detect significant associations.
3. The authors say that no circadian 24h rhythm was seen in liver but the liver. I think it is interesting that based on the Figure 3 a 12h rhythm can be seen in the liver, was there a bimodal 12h rhythm?
4. For analyses where pathways have been done, more informative would be to examine individual transcripts and give the test statistics for those.
5. Several studies have examined the circadian pattern of immune cell subsets, did the findings in this study align with previous findings?

Reviewer #3 (Remarks to the Author):

The authors report the discovery of cytosine modifications that oscillate with 24 hour circadian period in mice (*osc-modCs*). Using aged animals (9, 15 and 25 months) they also identify cytosine modifications that change with age (*age-modCs*). They demonstrate a statistically significant excess of *osc-modCs* among the *age-modCs* and provide arguments to support the hypothesis that circadian rhythmicity is causal for age-related changes in cytosine modification.

The finding of *osc-modCs* is apparently novel and would be of great interest. It is interesting that previous "dedicated studies" have failed to identify *osc-modCs*. What is different in this study that allowed detection where other studies failed?

The hypothesis that circadian oscillations are causal for the age related changes is likely to be controversial - very exciting if it holds up. However the evidence provided here is not experimentally driven. While I agree that the Hill causality criteria are met, these are not uncontroversial. The possibility of a shared common cause is hard to rule out. It is an exciting and bold hypothesis but perhaps the conclusions here could more qualified.

Some minor concerns:

Why are the studies conducted with technical replicates? Biological replicates are preferable and it is not clear how the mice (n=96 in total) are distributed across time points and age groups. Are mice of both sexes used in this study?

Circadian rhythms are affected by feeding time, most strongly in liver which undergoes a metabolic shift between fed and fasted states. How was feeding time controlled in this study?

There is a plethora of statistical tests (and p-values) reported in this manuscript. It is often hard to follow what are the units of the analysis used in these tests - sometimes mice, sometimes *modCs*? The extremely high significance levels reported are suspicious - and are likely a result of inappropriate tests embedded in standard enrichment analysis software. On the whole, I believe that the statistical methods applied are correct and appropriate but a summary of the different testing strategies applied would be helpful.

There is sufficient information provided for others to reproduce the work presented.

Reviewer #4 (Remarks to the Author):

In this study the authors measure the circadian behaviour of DNA methylation in lung and liver samples extracted from mice of different age. The mice were entrained by a 12h light / 12h dark

rhythm and sacrificed every two hours in order to obtain DNA methylation data at high temporal resolution. DNA methylation was measured by bisulphite padlock probes covering chromosome 7. The authors identify cytosines that exhibit a circadian methylation behaviour, which they call osc-modC. Oscillations of osc-modC correlated with mRNA oscillations of associated transcripts and osc-modCs were frequently enriched near E-box motifs. The authors further show that osc-modCs correlate with age-dependent changes in methylation. By measuring neutrophil DNA methylation from human blood samples the authors further identify that osc-modCs can in part explain variation in DNA methylation since these were highly enriched in regions with variable DNA methylation, i.e. enhancers and CGI shores.

In general this study provides further evidence that DNA methylation can vary in a circadian fashion and that numerous CpGs display such variation. The increased frequency of osc-modCs at sites with high methylation is interesting. These sites are known to coincide with TF binding sites. Unfortunately the authors do not explore this connection in more detail. That said, the presented study is purely correlative and the conclusions drawn from these correlations remain speculative and little is learned about the cause and consequences of circadian methylation patterns.

further comments:

The authors indicate that E-box motifs are frequently enriched near or at osc-modCs and that circadian DNA Methylation could regulate TF binding to these sites. However, the sites enriched (Fig 4c and d) do not contain CpGs. This argues against a direct role of methylation in regulating TF binding at these sites.

Some statements in the manuscript are exaggerated: i.e. the statement on the temporal relationship between DNA methylation and transcription is based Pearson's correlation values of 0.075 and 0.19. These small values are not very convincing and it would be good if the authors could show examples to support these data.

Furthermore, the circadian methylation dynamics are rather small and is always presented as average value or using other statistical measurements. I would prefer to see the actual methylation data showing these dynamics at genomic sites. Furthermore, given the small changes, the authors need to include information on the bisulphite conversion efficiency for all of their samples in order to exclude potential artifacts introduced by the treatment.

In figure 3a, the authors choose to calculate average methylation values over 1MB, and transform the data to a 0-1 range. This clearly exaggerates the results obtained from the direct methylation measurements. Here I suggest to show the direct DNA methylation values rather than the transformed values. I also do not understand why the authors choose such large windows of 1MB, which are rather unusual for representing DNA methylation obtained at nucleotide resolution.

Response to Reviewers' Comments

Reviewer #1:

- *The study of isolated neutrophils from a 52 year old man introduces a further level of complexity, and a range of technical questions about the likely effects of neutrophil isolation from other white cells on modC and transcription... I am not sure about the value of including the human leukocyte data. It fits uneasily with the liver and lung focus of the murine work and, although interesting, it raises a fresh set of questions rather than consolidating the main study.*

Isolation effect cannot create oscillations as the samples were processed as soon as possible after collection. Nevertheless, we agree with the reviewer that given the density of the current manuscript, removing this section could improve readability and coherence. We decided to take the reviewer's advice and removed the human neutrophil section from the manuscript.

- *Analytically, there is much to be admired in the paper, but the cost is that it makes it very demanding for the general reader. Harmonic regressions, double plotted data and glide reflections are examples of concepts that need explaining.*

A more lay description regarding harmonic regression (116-118), double plotted data (95-96 and 101), and glide reflections (221-223) were added to the manuscript.

- *The overlap between oscillation and age is not surprising, given that 50% of cytosines vary diurnally, and a statistical proof that this is not by chance is essential. The CpGs that vary with age in the human genome are well documented (for example by Horvath) and it would be good to know if these coincide with the murine loci.*

The statistical test used to measure the overlap between aging and oscillation cytosines (Fisher's exact test) takes the chance of overlap into account. Regarding Horvath's DNA modification "clock", the clock and epigenetic aging are two related but far from identical phenomena. He selected 353 CpGs which are best predictors of pan-tissue chronological age, but it is only a small fraction of all aging loci. It should be noted that the human component was removed from the manuscript based on this reviewer's suggestion. Our mouse experiments were specifically designed for uncovering the circadian - aging connection and tested the same set of CpGs in both temporal dimensions.

Reviewer #2:

- *The authors should emphasize the biological relevance of cytosine modifications also as regulators for gene expression and examine whether the osc-modCs defined in this paper are localized on known transcription factor binding sites. The authors even state on line 203 that "Methylation of E-box response elements can inhibit transcription factor binding, which suggests that osc-modCs may be involved in regulating the binding of transcription factors to E-box elements. In all, our findings show that osc-modCs are intricately linked to circadian transcriptomics." Formal analysis should be done to give an answer to this question as it is one of the key outcomes of the analysis.*

In order to investigate the possible role of transcription factor binding sites we performed motif analysis (See Methods - *Motif analysis*; line 598-603) using MEME and showed that E-box

elements were enriched near the sites that showed oscillations. Additionally, we further elaborated on the role of E-box elements in circadian regulation in the manuscript (205-213). We agree that a dedicated circadian DNA modification study specifically focused on transcription factor binding sites would be a better approach to addressing these questions.

- *N for the public data sets should be given in the text and figures to give the reader an understanding of the power to detect significant associations.*

Samples size information has been added in the revision (lines 185 and 290-291).

- *The authors say that no circadian 24h rhythm was seen in liver but the liver. I think it is interesting that based on the Figure 3 a 12h rhythm can be seen in the liver, was there a bimodal 12h rhythm?*

In fact, we did observe significant 24hr rhythm in the liver at a single nucleotide level. Chromosome-wide oscillations, however, were non-significant in the liver, and the following principal component analysis (lines 122-124) explains why - oscillating PC2 and PC3 are orthogonal. We did not find statistically significant individual cytosine- or chromosome-wide oscillations with a 12 hr period in any mouse tissue.

- *For analyses where pathways have been done, more informative would be to examine individual transcripts and give the test statistics for those.*

The transcriptomic data was from a previously published study which already detailed the test statistics of individual transcripts. However, we previously included two supplementary tables (Suppl. Table 14 and 15) detailing the properties of the transcripts that were used in our study for reproducibility.

- *Several studies have examined the circadian pattern of immune cell subsets, did the findings in this study align with previous findings?*

Our findings were consistent with previous studies that observed oscillations of leukocyte subfractions, which was referenced in our previous version of the manuscript (former Suppl. Figure 9). In the new revision, the human component was removed as suggested by the first reviewer.

Reviewer #3:

- *The finding of osc-modCs is apparently novel and would be of great interest. It is interesting that previous "dedicated studies" have failed to identify osc-modCs. What is different in this study that allowed detection where other studies failed?*

Compared to the previous attempts to identify circadian DNA modifications (Vollmers et al, *Cell Metab*, 16(6), 833-845 (2012) ; Azzi et al, *Nat Neurosci*, 17(3), 377-382 (2014)), there are numerous advancements made in our study. For instance, we used a deep (~1,800X) bisulfite sequencing method using padlock probes, which is significantly more sensitive than the low resolution MeDIP (Azzi et al, 2014) or shallow (~5X) whole genome bisulfite sequencing (Vollmers et al, 2012). The studies also differed in power; for example both of the previous studies reduced the circadian cycle to 2 time points and attempted to identify DNA methylation differences. In our study, we meticulously tested 3 mice every 2 hours (12 time points) over a 24

hr period. To reduce experimental noise, we also tested each biological sample in technical triplicate.

- *The hypothesis that circadian oscillations are causal for the age related changes is likely to be controversial - very exciting if it holds up. However the evidence provided here is not experimentally driven. While I agree that the Hill causality criteria are met, these are not uncontroversial. The possibility of a shared common cause is hard to rule out. It is an exciting and bold hypothesis but perhaps the conclusions here could more qualified.*

We agree with the reviewer's comment that although it is bold and exciting, it still requires additional research. Therefore, we have made adjustments in our Results section (lines 285-287) to suggesting that additional experiments may be required to concretely establish causality.

- *Why are the studies conducted with technical replicates? Biological replicates are preferable and it is not clear how the mice (n=96 in total) are distributed across time points and age groups. Are mice mice of both sexes used in this study?*

Each time point (ZT) was measured in three biological and three technical replicates, therefore each ZT has 9 measurements (total n = 756 of tested DNA samples; this includes all biological and technical replicates for mapping of modified, methylated, and hydroxymethylated cytosines). The technical replicates were used to increase statistical power by reducing technical variation, filter signals by comparing technical variation with biological differences, and as a means to be robust against outlier samples. Only males were used in this study, and it is now noted in the manuscript (lines 66, 84, 386).

- *Circadian rhythms are affected by feeding time, most strongly in liver which undergoes a metabolic shift between fed and fasted states. How was feeding time controlled in this study?*

Food was available *ad libitum*. Actographic data showed that the animals were usually active during lights off and dormant during lights on, suggesting that most feeding took place during physically active times.

- *There is a plethora of statistical tests (and p-values) reported in this manuscript. It is often hard to follow what are the units of the analysis used in these test - sometimes mice, sometimes modCs? The extremely high significance levels reported are suspicious - and are likely a result of inappropriate tests embedded in standard enrichment analysis software. On the whole, I believe that the statistical methods applied are correct and appropriate but a summary of the different testing strategies applied would be helpful. There is sufficient information provided for others to reproduce the work presented.*

We are very glad to learn that the reviewer thinks we used appropriate statistical tests and that we provided sufficient information for replication studies. We believe we provided exhaustive details in the Methods section necessary for understanding of the rationale and usage of specific statistical tool as well as replication of our findings.

Reviewer #4:

- *In general this study provides further evidence that DNA methylation can vary in a circadian fashion and that numerous CpGs display such variation. The increased frequency of osc-modCs at sites with high methylation is interesting. These sites are known to coincide with TF binding sites. Unfortunately the authors do not explore this connection in more detail. That said, the presented study is purely correlative and the conclusions drawn from these correlations remain speculative and little is learned about the cause and consequences of circadian methylation patterns.*

Our primary goal was to document the circadian rhythm in cytosine modification and demonstrate that cytosine modification is a *bona fide* member of circadian machinery, altogether with transcriptome, proteome, metabolome, and other elements of epigenome (such as histone modification). As we mentioned in the introduction, we were not the first to investigate circadian DNA modification effects. At least two other comprehensive studies were attempted but failed (see related response to reviewer 3). It required significant technological expertise, high throughput sequencing technology for fine mapping of modified cytosines, and very meticulous study design to accomplish the goal of identification of osc-modCs. Our secondary goal was to initiate investigations beyond circadian epigenome *per se*, such as relationship between circadian oscillations of epigenome and transcriptome and aging. We cannot address all important questions in one study (which has already been commented as dense and hard to read) but - along the lines of the reviewer's comment - we do think that further research effort should be dedicated to addressing mechanistic questions surrounding circadian DNA modification.

- *The authors indicate that E-box motifs are frequently enriched near or at osc-modCs and that circadian DNA Methylation could regulate TF binding to these sites. However, the sites enriched (Fig 4c and d) do not contain CpGs. This argues against a direct role of methylation in regulating TF binding at these sites.*

Yes, the osc-modCs are proximal to the E-box motif sites (± 100 bp). E-box binding proteins form large complexes that can interact with DNA methyltransferases, and we speculated that osc-modCs formed outside of the motif is related to this. We have included this hypothesis in the manuscript (lines 205-213).

- *Some statements in the manuscript are exaggerated: i.e. the statement on the temporal relationship between DNA methylation and transcription is based Pearson's correlation values of 0.075 and 0.19. These small values are not very convincing and it would be good if the authors could show examples to support these data.*

We were very cautious to structure a robust hypothesis test to make sure that this observation was not due to a chance event (e.g., we only considered each transcript once and averaged all cytosine modification data in the gene region). We do not believe example transcripts would aid in the interpretation of the presented evidence, but would instead bias the view of the data. The documented significant phase relationship between transcript oscillations and cytosine modification levels serves as secondary evidence, both supporting a systematic connection between DNA modification and transcriptional oscillations. Furthermore, it should be noted that the public dataset for transcriptomic data had a different study design, used different strain of mice, and were of different ages; all of which can reduce correlations.

- *Furthermore, the circadian methylation dynamics are rather small and is always presented as average value or using other statistical measurements. I would prefer to see the actual methylation data showing these dynamics at genomic sites.*

We have included a new supplementary figure to show some representative examples of CpGs with good or no oscillation (Supplementary Figure 2).

- *Furthermore, given the small changes, the authors need to include information on the bisulphite conversion efficiency for all of their samples in order to exclude potential artifacts introduced by the treatment.*

Using CpH methylation as proxy for bisulfite conversion efficiency for each sample, we did not find any evidence for oscillation based on the bisulfite conversion rate. The mean \pm SD of the CpH for the samples used in this study was $0.6 \pm 0.6\%$, which also indicates very good conversion efficiency. This information has been added to the Methods (line 418-419).

- *In figure 3a, the authors choose to calculate average methylation values over 1MB, and transform the data to a 0-1 range. This clearly exaggerates the results obtained from the direct methylation measurements. Here I suggest to show the direct DNA methylation values rather than the transformed values. I also do not understand why the authors choose such large windows of 1MB, which are rather unusual for representing DNA methylation obtained at nucleotide resolution.*

In this figure, we wanted to illustrate that circadian cytosine modification can be a chromosome-wide phenomenon. We believe that our 1Mb bin is a good compromise between resolution and generalization given the scale of the data. Scaling provides an even visual field so that oscillations between varying average densities of cytosine modification (i.e. from 0%-100%) become comparable. Relatedly, a plot of the direct DNA methylation values is not suitable for a generalized chromosome-wide view because different genomic elements exhibit major differences in terms of density of modified cytosines.

Reviewers' comments:

Reviewer #1 (Remarks to the Author):

This paper is greatly improved. I found it much easier to understand what was going on, and there is now a clear narrative to interesting results.

I appreciate the removal of jargon and the explanation of the more esoteric analyses.

Reviewer #2 (Remarks to the Author):

The authors have answered to all the points raised, and revised the manuscript accordingly. I do not have any further comments.

Reviewer #3 (Remarks to the Author):

Thanks for clarify the question about technical and biological replicates. In general (i.e. always) biological replicates are most effective at reducing variation. However technical replicates are helpful provided that they are analyzed correctly. In this situation - where the number of technical replicates is equal across all biological samples, the simplest thing to do is average the technical values and analyze the data on biological units. This should be reflected in the degrees of freedom in the regression. Alternatively one could fixed hierarchical (i.e., mixed effects) models.

This brings me to my second point and an apology for the unfortunate typo in my previous comments. Allow me to correct and rephrase the comment:

I believe - but I am not yet convinced - that the statistical methods applied in the analysis of these data are correct. There is not sufficient information provided for others to reproduce the statistical tests as presented. Additional details in the methods are needed to address this concern.

I would like to see further clarification about the effects of feeding on c-Mods in liver. My question centers around whether feeding behavior changes with age - if this is the case then the circadian changes in methylation with age could simply reflect changes in feeding behavior with age. Is there any evidence to support that feeding is consistent across age? Also, how were mice handled prior to euthanasia - were they transferred directly from the home cage, or were they separated from food for a fixed period prior?

Reviewer #4 (Remarks to the Author):

The authors have addressed most of my comments, but there are still some open points that need to be clarified.

Related to the E-box motifs. The authors claim to detect E-box motifs around *osc*-mods that could be directly or indirectly regulated by DNA methylation. I think these are wild speculations that should be completely omitted unless there is experimental evidence provided in this manuscript. The newly added speculations with Myc/Max and DNMT3A are not convincing.

The authors still do not show any examples to support their claims about temporal relationship between methylation and transcription. If their hypothesis test is as robust as they claim in the rebuttal, then such correlations should be also observed at a single gene level. I think such examples are needed to convince the readers that the minor correlations from this comparison are indeed reflected by direct measurements, and not due solely based on statistical measurements which in genomic studies are always significant due to the high statistical power.

Also if the authors mention that the public transcriptome data has a different design which would not allow such comparisons, why was it used to calculate this temporal relationship at all?

Response to Reviewers' Comments V2

Reviewer #3:

- *Thanks for clarify the question about technical and biological replicates. In general (i.e. always) biological replicates are most effective at reducing variation. However technical replicates are helpful provided that they are analyzed correctly. In this situation - where the number of technical replicates is equal across all biological samples, the simplest thing to do is average the technical values and analyze the data on biological units. This should be reflected in the degrees of freedom in the regression. Alternatively one could fixed hierarchical (i.e., mixed effects) models.*

We would like to note that, in addition to technical replicates, we also used biological replicates. Each ZT in our study was represented by 2-3 independent animals for every age group. We did take the median of the three technical replicates for our analysis, and this has been clarified in the current version of the manuscript (lines 472-479). The mixed-effects model was tested initially but was abandoned because the p-value histogram using random data resulted in a non-uniform distribution, which would lead to problems interpreting the results.

- *This brings me to my second point and an apology for the unfortunate typo in my previous comments. Allow me to correct and rephrase the comment: I believe - but I am not yet convinced - that the statistical methods applied in the analysis of these data are correct. There is not sufficient information provided for others to reproduce the statistical tests as presented. Additional details in the methods are needed to address this concern.*

We have added much more detail regarding the statistical methods in the methods section (lines 472-581)

- *I would like to see further clarification about the effects of feeding on c-Mods in liver. My question centers around whether feeding behavior changes with age - if this is the case then the circadian changes in methylation with age could simply reflect changes in feeding behavior with age. Is there any evidence to support that feeding is consistent across age? Also, how were mice handled prior to euthanasia - were they transferred directly from the home cage, or were they separated from food for a fixed period prior?*

A previous study showed that in C57BL/6 mice *ad libitum* food consumption does not change as a function of age from 4 - 16 months (Ferguson et al, *Effect of long-term caloric restriction on oxygen consumption and body temperature in two different strains of mice*. Mech Ageing Dev 128, 539-545, (2007)). Interestingly, weight of these animals steadily increased with age, likely due to alterations in their metabolism.

The animals were individually transported to a different room just prior to euthanizing. We did not interact with the animals (i.e. separate the animals from food) prior to the experiment because this may stress out the animals and affect the normal circadian rhythm.

Reviewer #4:

- *Related to the E-box motifs. The authors claim to detect E-box motifs around osc-mods that could be directly or indirectly regulated by DNA methylation. I think these are wild speculations that should be completely omitted unless there is experimental evidence provided in this manuscript. The newly added speculations with Myc/Max and DNMT3A are not convincing.*

Following the reviewer's recommendation, we removed any claims regarding E-box regulation via cytosine modification. However, we decided to keep the *Myc/Max* and *DNMT3A* statement that addressed (and satisfied) reviewer #2's concerns regarding additional biological interpretation in the previous round of review.

- *The authors still do not show any examples to support their claims about temporal relationship between methylation and transcription. If their hypothesis test is as robust as they claim in the rebuttal, then such correlations should be also observed at a single gene level. I think such examples are needed to convince the readers that the minor correlations from this comparison are indeed reflected by direct measurements, and not due solely based on statistical measurements which in genomic studies are always significant due to the high statistical power.*

As suggested by the reviewer, we have chosen several examples of gene level data to illustrate the relationship (Supplementary Fig. 7; lines 182-185).

- *Also if the authors mention that the public transcriptome data has a different design which would not allow such comparisons, why was it used to calculate this temporal relationship at all?*

Our study is focused on the circadian cytosine modifications, which was performed at an unprecedented scale and precision. While we had no plans to perform a dedicated circadian-aging transcriptomic study, the publicly available transcriptomic datasets, although not perfect, provided new insights and corroborated our epigenomic findings.

REVIEWERS' COMMENTS:

Reviewer #3 (Remarks to the Author):

I appreciate the author's efforts to address my comments.

Reviewer #4 (Remarks to the Author):

No further comments.